# Modulation Signal Recognition of Underwater Acoustic Communication Based on Archimedes Optimization Algorithm and Random Forest

**DOI:** 10.3390/s23052764

**Published:** 2023-03-02

**Authors:** Maofa Wang, Zhenjing Zhu, Gaofeng Qian

**Affiliations:** 1School of Mechanical Engineering, Hangzhou Dianzi University, Hangzhou 310018, China; 2Marine Technology and Equipment Research Center, Hangzhou Dianzi University, Hangzhou 310012, China

**Keywords:** modulation recognition, random forest, Archimedes optimization algorithm, underwater acoustic communication

## Abstract

This paper researches the recognition of modulation signals in underwater acoustic communication, which is the fundamental prerequisite for achieving noncooperative underwater communication. In order to improve the accuracy of signal modulation mode recognition and the recognition effects of traditional signal classifiers, the article proposes a classifier based on the Archimedes Optimization Algorithm (AOA) and Random Forest (RF). Seven different types of signals are selected as recognition targets, and 11 feature parameters are extracted from them. The decision tree and depth obtained by the AOA algorithm are calculated, and the optimized random forest after the AOA algorithm is used as the classifier to achieve the recognition of underwater acoustic communication signal modulation mode. Simulation experiments show that when the signal-to-noise ratio (SNR) is higher than −5dB, the recognition accuracy of the algorithm can reach 95%. The proposed method is compared with other classification and recognition methods, and the results show that the proposed method can ensure high recognition accuracy and stability.

## 1. Introduction

The ocean is an important part of the territory of each country, with abundant fisheries, energy, and mineral resources [1]. For all countries, rapid access to ocean information is crucial for gaining international initiative [2]. In addition, in maritime warfare, it is important to quickly understand the battlefield situation and obtain military intelligence [3], thus making underwater communication technology and systems particularly important. The acoustic modem is a core component of underwater communication systems, responsible for converting digital data into acoustic signals and sending them to underwater receivers while also demodulating received acoustic signals and converting them into digital data [3,4]. Signal modulation recognition is an indispensable part of underwater communication which has become a research focus for various countries in recent years [5,6,7,8].

Compared with other communication channels, the underwater acoustic channel has a more complex transmission environment [9,10]. The propagation speed and available bandwidth of underwater acoustic communication signals are both limited. In addition, the underwater acoustic channel also has the characteristics of time-varying multipath. The transmitter and receiver of the signal undergo relative motion due to the flow of seawater, which can distort the waveform of the received signal. This greatly affects the robustness of signal transmission in the ocean, which may lead to a decrease in the decoding performance of underwater acoustic communication systems or even cause them to fail to work properly [11]. Therefore, the recognition of underwater acoustic communication signal modulation methods poses certain challenges.

Water acoustic signal modulation recognition has two main methods: manual recognition and machine recognition. In the early days, manual recognition relied mainly on observing the time domain and frequency domain characteristics of the signal to determine its modulation method [12]. However, as the modulation methods gradually became more complex, manual recognition often resulted in errors or even failed recognition. With the continuous development of statistical theory, many decision theories for signal modulation recognition have emerged, such as the Average Likelihood Ratio Test (ALRT) [13], the Generalized Likelihood Ratio Test (GLRT) [14], and the Hybrid Likelihood Ratio Test (HLRT) [15]. Xu et al. [16] summarized the likelihood ratio test methods in the automatic modulation recognition process. Two sets of different simulated data were used for comparison and verification. The simulation results show that the performance of the ALRT is better than other methods, especially showing better robustness at high SNR. Wu et al. [17] studied the problem of recognizing the APSK modulation method in DVB-S2 and proposed a radius constellation map-based APSK modulation recognition method based on the GLRT. The simulation results show that this method is effective in recognizing the APSK modulation under frequency offset. Hameed et al. [18] investigated the application of HLRT in linear digital modulation classification. The study found that HLRT has high complexity and is not easily applicable. However, there are issues with the above research, such as high computational cost and poor universality, making it difficult to achieve real-time intelligent identification of modulation schemes.

In order to address the above problems, feature-based modulation recognition technology has gradually developed in underwater acoustics engineering, with the key being the distinction of signal features and the extraction of feature parameters. Commonly used feature parameters include: instantaneous statistical features [12], higher-order cumulants [19], cyclic spectral features [20], autocorrelation features [21]; constellation diagram features [22]; and wavelet transform features [23]. In recent years, with the continuous development of Artificial Intelligence (AI) and machine learning technology, algorithms, such as Support Vector Machine (SVM) [24], Neural Network [25], Machine Learning [26,27], Decision Tree (DT) [28], and RF [29], have emerged in the field of intelligent classification recognition. B. Kim et al. [30] used 19 statistical features of signals as inputs to a four-layer neural network classifier for the classification and recognition of BPSK, QPSK, 8PSK, 16QAM, and 64QAM signals. Zhou et al. [31] represented the signal features using four instantaneous parameters and four statistical parameters and applied a Genetic Algorithm to design a Backpropagation Neural Network (BPNN) automatic modulation classifier. Simulation results showed that the recognition accuracy was 94.5% when SNR was 3.0. Meng et al. [32] proposed a cascade classifier by combining an SVM and a Convolutional Neural Network for radar signal background. This classifier used multiple decision thresholds for decision-making, with each threshold corresponding to a different classifier, and the cascade structure could be used to reduce the false alarm rate and the missed detection rate. Simulation results showed that the recognition accuracy was 82.27% when SNR was −2.0. Wei et al. [33] proposed a novel signal classification method using SVM based on hybrid features, including cyclostationary and information entropy. The method combines the theories of cyclostationary and entropy and uses a one-against-one SVM as a classifier. Simulation results show that in the Additive White Gaussian Noise (AWGN) channel environment, the recognition accuracy is 85.92% when SNR is 0. Hassan et al. [34] proposed a method for automatic modulation recognition using wavelet transform and neural networks. The authors used wavelet transform to convert the modulated signals into the wavelet domain and extract the energy features of wavelet coefficients. Then, they used neural networks for modulation recognition. Experimental results show that this method can effectively recognize modulated signals under various SNR conditions.

These AI-based classification and recognition methods have good performances even in low SNR environments. However, these methods often require a large amount of training data and are prone to overfitting, resulting in low model stability and making them difficult to apply in practical engineering. Building upon the aforementioned research, this paper proposes a signal classification and recognition method based on the AOA and RF algorithms. By optimizing the two key parameters of the number of decision trees and the maximum depth of decision trees, this method aims to provide various optimization solutions for engineers to address practical engineering problems.

This paper is organized as follows. Section 2 presents the theoretical background of the AOA algorithm and the basic structure of the RF algorithm. Section 3 describes the extraction process of feature parameters of the modulation signal. Section 4 gives the classification results and compares them with other methods. The work performed for this study and the subsequent work are discussed in Section 5.

## 2. Methodology

### 2.1. Archimedes Optimization Algorithm

#### 2.1.1. Archimedes Principle

The Archimedes principle states that when an object is partially or completely immersed in a fluid, it is subject to a buoyant force exerted by the fluid. If the magnitude of the buoyant force is equal to the weight of the fluid displaced by the object, then the object is considered to be in equilibrium. If the magnitude of the buoyant force is not equal to the weight of the displaced fluid, the object is not in equilibrium. Each object immersed in a fluid has a tendency to reach equilibrium, and because each object has a different density ρ and volume v, the acceleration a toward equilibrium is different for each object.

When an object is immersed in a fluid, it is in equilibrium, as shown in Formulas (1).
(1)$$\begin{array}{l} F_{b}=W_{o}\\ \rho_{b}v_{b}a_{b}=\rho_{o}v_{o}a_{o} \end{array}$$

In the formulas, $F_{b}$ is the buoyancy of the object, and $W_{o}$ is the gravity of the fluid displaced by the object. Depending on the force applied and the force received, subscript b denotes the fluid to which the force is applied, and subscript o denotes the object to which the force is applied when immersed in the fluid. This leads to the conclusion that Formula (2) is the acceleration expression.
(2)$$a_{0}=\frac{\rho_{b}v_{b}a_{b}}{\rho_{o}v_{o}}$$

The expression for the equilibrium state at this moment when an object submerged in water collides with a neighboring object is Formula (3).
(3)$$\begin{array}{l} F_{b}=W_{o}+W_{r}\\ \rho_{b}v_{b}a_{b}=\rho_{o}v_{o}a_{o}+\rho_{r}v_{r}a_{r} \end{array}$$

#### 2.1.2. Initialization

In 2020, Hashim et al. [35] proposed an AOA optimization algorithm, which is based on the Archimedes principle and, similar to other metaheuristic algorithms, is a population-based optimization algorithm. The process mainly consists of an initialization phase, a global exploration phase, and a local exploitation phase. In the initialization phase, the density (*den*), volume (*vol*), and acceleration (*acc*) of the object are randomly assigned. To speed up the iteration process, the AOA algorithm evaluates the initial population in this process, selects the best individual ($x_{best}$), and determines the optimal density ($den_{best}$), volume ($vol_{best}$), and acceleration ($acc_{best}$).

Formula (4) illustrates how an iterative process works.
(4)$$\begin{array}{l} den_{i}^{t+1}=den_{i}^{t}+rand\times (den_{best}-den_{i}^{t})\\ v_{i}^{t+1}=vol_{i}^{t}+rand\times (vol_{best}-vol_{i}^{t}) \end{array}$$

In the formulas, X is a vector with dimension D, and its value is between 0 and 1.

#### 2.1.3. Transfer Operator

By influencing the value of Transfer Factors (TF), the algorithm switches individuals between collision and equilibrium states, i.e., from the global exploration phase to the local development phase, in order to obtain specific solutions to the optimization problem.

In Formula (5), the definition of TF is displayed.
(5)$$TF=\exp(\frac{t-t_{\max}}{t_{\max}})$$

In the formulas, $t_{\max}$ is the maximum number of iterations of the algorithm, and *t* is the current number of iterations. TF≤0.5 stands for collision between objects, and the algorithm is in the global exploration stage. If TF>0.5 represents that there is no collision between objects, the algorithm is in the local development stage.

#### 2.1.4. Object’s Acceleration

In different stages, there are different acceleration formulas and object position formulas. In the global exploration stage, the *acc* expression of the object is shown in Formula (6).
(6)$$acc_{i}^{t+1}=\frac{den_{mr}+vol_{mr}+acc_{mr}}{den_{i}^{t+1}\times vol_{i}^{t+1}}$$

In the formulas, $acc_{mr},den_{mr}$, and $vol_{mr}$ are the acceleration, density, and volume of a random individual in the iterative process.

In the local development stage, the *acc* expression of the object is shown in Formula (7).
(7)$$acc_{i}^{t+1}=\frac{den_{\mathrm{best}}+vol_{\mathrm{best}}+acc_{\mathrm{best}}}{den_{i}^{t+1}\times vol_{i}^{t+1}}$$

To balance the relationship between the global exploration phase and the local development phase, it is necessary to normalize the *acc* of each object. The normalization formula for *acc* is shown in Formula (8).
(8)$$acc_{i-norm}^{t+1}=u\times \frac{acc_{i}^{t+1}+\min(acc)}{\max(acc)\times \min(acc)}+l$$

In the formulas, the normalized *acc* is expressed as a percentage. The values of *u* and *l* are set to 0.9 and 0.1, respectively.

#### 2.1.5. Object’s Position

The following equation can be used to calculate the object’s location during the global exploration stage:(9)$$x_{i}^{t+1}=x_{i}^{t}+c_{1}\times rand\times acc_{i-norm}^{t+1}\times d\times \left(x_{\mathrm{rand}}-x_{i}^{t}\right)$$

In the formulas, $C_{1}$ is a constant with a value equal to 2; *d* is the density factor, and its calculation formula is as follows:(10)$$d^{t+1}=\exp\left(\frac{t_{\max}-t}{t_{\max}}\right)-\left(\frac{t}{t_{\max}}\right)$$

In the local development stage, the formula for determining the position of the object is as follows:(11)$$x_{i}^{t+1}=x_{best}^{t}+F\times C_{2}\times rand\times acc_{i-norm}^{t+1}\times d\times (T\times x_{best}-x_{i}^{t})$$

In the formulas, $C_{2}$ is a constant with a value equal to 6. $T=C_{3}\times TF$ and $T\in \left[0.3C_{3},1\right]$; $C_{3}$ is a constant. The parameter F is used to change the direction of object movement, which is defined as follows:(12)$$F=\{\begin{array}{cc} +1, & p\leq 0.5\\ -1, & p>0.5 \end{array}$$

In the formulas, $P=2rand-C_{4}$, and $C_{4}$ is a constant.

In summary, the flowchart of the AOA algorithm is shown in Figure 1.

### 2.2. Random Forest

RF, also known as random decision forest, is an ensemble learning algorithm that can effectively avoid overfitting. The algorithm consists of multiple DTs. In the classification process, a subset of the training set is randomly selected with replacement to construct a subset, and each subset corresponds to a DT. The DT are then trained independently to obtain separate training results. Each DT is independent and does not interfere with each other. The final decision is determined by the classifier based on the voting results of each DT.

RF, compared to a single DT, has better classification performance and overcomes the overfitting problem during the training process, with strong generalization ability. The basic model of a RF is shown in Figure 2.

#### 2.2.1. Determine the Number of DT

In the RF algorithm, the number *N* of DT is related to the input dataset, and the number of DT has a significant impact on the classification accuracy of the random forest algorithm. When the number is too small, the classification accuracy will decrease. When the number is too large, the computational complexity of the model will increase, and the training time will become too long.

This paper investigates in depth the correspondence between the number of DT and the dataset and, through continuous simulation experiments, ultimately selects a random forest with *N* = 100 DT for specific modulation signal classification recognition, achieving a balance between computation time and classification accuracy.

#### 2.2.2. Determine the Depth of the DT

The minimum *Gini* index is the guiding principle for node splitting, and its calculation is as follows:(13)$$Gini(S)=1-\overset{m}{\underset{i=1}{\sum }}P_{i}^{2}$$

In the formulas, $P_{i}$ is the probability that category $C_{i}$ appears in the training dataset S. The divided *Gini* coefficient is as follows if the RF at this time consists of *N* DT:(14)$$Gini_{split\;}(S)=\frac{1}{\;S}\overset{k}{\underset{n=1}{\sum }}\left|S_{n}\right|Gini\left(S_{n}\right)$$

### 2.3. Overall Approach of this Study

The overall technical roadmap of the modulation signal recognition method proposed in this paper is shown in Figure 3. This method mainly consists of two parts: optimizing the number and depth of DT using the AOA algorithm and using the RF for modulation signal classification and recognition.

The specific steps are as follows: First, preprocess the signal and generate training and testing samples based on the signal’s feature parameters. Then, input the DT number and depth obtained by the AOA algorithm into the RF. Finally, after training with the RF, obtain the classification results of the modulated signals.

## 3. Feature Parameter Extraction and Analysis

### 3.1. Feature Parameters Based on Instantaneous Information

The communication modulation signals have varying degrees of differences in amplitude, frequency, and phase, which is the key to distinguishing different communication modulation signals. The extraction of instantaneous information from signals is the premise and foundation for realizing signal recognition and classification based on instantaneous features. Instantaneous features are estimated using the Hilbert transform, and the signal X(n) is first transformed using the Hilbert transform as follows:(15)X(t)=S(t)+jHilbert[S(t)]

The signal’s instantaneous amplitude a(n), instantaneous phase ϕ(n), and instantaneous frequency f(n) are, in turn, as follows:(16)$$\begin{array}{l} a(t)=\sqrt{S^{2}(t)+Hilbert^{2}[S(t)]}\\ \phi (t)=\tan^{-1}\left(\frac{Hilbert[S(t)]}{S(t)}\right)\\ f(t)=\frac{1}{2\pi }\frac{d\varphi (t)}{dt} \end{array}$$

Based on instantaneous amplitude a(n), instantaneous phase ϕ(n), and instantaneous frequency f(n), the following four feature parameters are extracted.

#### 3.1.1. Absolute Amplitude Standard Deviation $\sigma_{aa}$

The standard deviation of the absolute value of the normalized zero center instantaneous amplitude is $\sigma_{aa}$, and its calculation formula is as follows:(17)$$\sigma_{aa}=\sqrt{\frac{1}{c}\left(\sum_{a_{n}(i)>a_{t}}a_{NL}^{2}(i)\right)-\frac{1}{c}{\left(\sum_{a_{n}(i)>a_{t}}\left|a_{NL}(i)\right|\right)}^{2}}$$

In the formulas, $a_{t}$ is the threshold level of amplitude decision, which is used to judge whether it is a weak signal segment. When the signal meets a certain threshold level, it is a non-weak signal; *c* is the number of non-weak signals in all adopted data $N_{i}$. The $a_{NL}(i)$ is the nonlinear component of instantaneous amplitude at zero center when the carriers are completely synchronized $a_{NL}(i)=a(i)-a_{0}$, where $a_{0}=\frac{1}{N_{s}}\sum_{i=1}^{N_{s}}a(i)$, and a(i) is the instantaneous amplitude.

#### 3.1.2. Absolute Frequency Standard Deviation $\sigma_{af}$

The standard deviation of the absolute value of the normalized zero center instantaneous frequency is $\sigma_{af}$, and its calculation formula is as follows:(18)$$\sigma_{af}=\sqrt{\frac{1}{c}\left(\underset{a_{n}(i)>a_{t}}{\sum }f_{NL}^{2}(i)\right)-\frac{1}{c}{\left(\underset{a_{n}(i)>a_{t}}{\sum }\left|f_{NL}(i)\right|\right)}^{2}}$$

In the formulas, f(i) is instantaneous frequency.

#### 3.1.3. Absolute Phase Standard Deviation $\sigma_{ap}$

The standard deviation of the absolute value of the instantaneous phase nonlinear component of a non-zero center non-weak signal is $\sigma_{ap}$, and its calculation formula is as follows:(19)$$\sigma_{ap}=\sqrt{\frac{1}{c}\left(\sum_{a_{n}(i)>a_{t}}\phi_{NL}^{2}(i)\right)-\frac{1}{c}{\left(\sum_{a_{n}(i)>a_{t}}\left|\phi_{NL}(i)\right|\right)}^{2}}$$

In the formulas, φ(i) is an instantaneous phase.

#### 3.1.4. Direct Phase Standard Deviation $\sigma_{dp}$

The standard deviation of the instantaneous phase nonlinear component of the zero-center non-weak signal is the direct phase standard deviation $\sigma_{dp}$. The difference between the direct phase standard deviation $\sigma_{dp}$ and the absolute phase standard deviation $\sigma_{ap}$ lies in whether the absolute value is taken in the calculation. The direct phase standard deviation $\sigma_{dp}$ can reflect the features of direct phase information existing in the signal. Its calculation formula is as follows:(20)$$\sigma_{dp}=\sqrt{\frac{1}{c}\left(\sum_{a_{n}(i)>a_{t}}\phi_{NL}^{2}(i)\right)-\frac{1}{c}{\left(\sum_{a_{n}(i)>a_{t}}\phi_{NL}(i)\right)}^{2}}$$

### 3.2. Feature Parameters Based on Higher-Order Cumulant

Since the moments above the second order of Gaussian random variables do not provide new information, the cumulant can be obtained by subtracting the higher-order moments of adjacent orders, and the higher-order cumulant can reduce the influence of white noise on the modulation signal under Gaussian distribution. However, other parameters in the signal will produce different high-order accumulated information after different-order operations, which can also be used to classify and identify the signal.

Let x(n) be a k-order stochastic process with zero mean, and the k-order moments and k-order cumulants are, respectively, as follows:(21)$$\begin{array}{l} M_{kx}(l_{1},\ldots ,l_{k-1})=Mom\left[x(n),x(n+l_{1}),\ldots x(n+l_{k-1})\right]\\ C_{kx}(l_{1},\ldots ,l_{k-1})=Cum\left[x(n),x(n+l_{1}),\ldots x(n+l_{k-1})\right] \end{array}$$

In the formulas, Mom(·) and Cum(·) are joint moments and joint cumulants, respectively.

In practice, the higher-order cumulant of the signal cannot be directly obtained by calculation, so it is necessary to estimate the cumulant from limited data.

Let the received actual signal be $r_{k}$, k=1,……,n the cumulative quantity of each order is as follows:(22)$$\begin{array}{l} {\hat{C}}_{20}={\hat{M}}_{20}=\frac{1}{N}\sum_{k=1}^{N}r_{k}^{2}\\ {\hat{C}}_{21}={\hat{M}}_{21}=\frac{1}{N}\sum_{k=1}^{N}{\left|r_{k}\right|}^{2}\\ {\hat{C}}_{40}={\hat{M}}_{40}-3{\hat{M}}_{20}=\frac{1}{N}\sum_{k=1}^{N}r_{k}^{4}-3{\hat{M}}_{20}\\ {\hat{C}}_{41}={\hat{M}}_{41}-3{\hat{M}}_{21}{\hat{M}}_{20}=\frac{1}{N}\sum_{k=1}^{N}r_{k}^{3}r_{k}^{*}-3{\hat{M}}_{21}{\hat{M}}_{20}\\ {\hat{C}}_{42}={\hat{M}}_{42}-{\left|{\hat{M}}_{20}\right|}^{2}-2{\hat{M}}_{21}^{2}=\frac{1}{N}\sum_{k=1}^{N}{\left|r\right|}_{k}^{4}-{\left|{\hat{M}}_{20}\right|}^{2}-2{\hat{M}}_{21}^{2}\\ {\hat{C}}_{60}={\hat{M}}_{60}-15{\hat{M}}_{40}{\hat{M}}_{20}+30{\hat{M}}_{20}^{2}=\frac{1}{N}\sum_{k=1}^{N}r_{k}^{6}-15{\hat{M}}_{40}{\hat{M}}_{20}+30{\hat{M}}_{20}^{2}\\ {\hat{C}}_{63}={\hat{M}}_{63}-6{\hat{M}}_{41}{\hat{M}}_{20}-9{\hat{M}}_{42}{\hat{M}}_{21}{}^{2}+18{\hat{M}}_{20}{}^{2}{\hat{M}}_{21}+12{\hat{M}}_{21}{}^{3} \end{array}$$

To better identify and classify various modulation signals and avoid situations where a single cumulative quantity may not effectively distinguish between signal types, this paper uses various combinations of cumulative quantities of different orders as recognition feature labels, and constructs feature parameters $F_{i}$ using combinations of second, fourth, sixth, and eighth order cumulative quantities. The definitions of the four types of feature parameters constructed as follows:(23)$$F1=\frac{\left|C_{40}\right|}{\left|C_{42}\right|}\;F2=\frac{\left|C_{41}\right|}{\left|C_{42}\right|}\;F3=\frac{\left|C_{42}\right|}{\left|C_{21}\right|}\;F4=\frac{{\left|C_{63}\right|}^{2}}{{\left|C_{42}\right|}^{3}}$$

To eliminate the influence of phase and signal energy, the obtained feature parameters are taken as absolute values, and the ratio method is used to normalize the accumulated results.

According to the above scheme, the feature parameters $F_{i}$ of six modulation signals, 2ASK, 4ASK, 2FSK, 4FSK, 2PSK, and 4PSK, were calculated, and the theoretical values of the feature parameters for each signal are shown in Table 1.

From Table 1, it can be seen that the theoretical value of the feature parameter $F_{1}$ is 1 for MASK and MPSK, while it is 0 for MFSK. Therefore, the feature parameter $F_{1}$ can be used to distinguish MFSK from other types of signals. Similarly, the feature parameter $F_{2}$ can be used to divide signal types into two groups: [2FSK, 4FSK, 4PSK] and [2ASK, 4ASK, 2PSK]. The feature parameter $F_{3}$ can be used to divide signal types into two groups: [2ASK, 2PSK] and [4ASK, 2FSK, 4FSK, 4PSK]. The feature parameter $F_{4}$ can be used to divide signal types into two groups: [2ASK, 4ASK, 2PSK] and [2FSK, 4FSK, 4PSK].

It is worth noting that high-order cumulants are not effective in distinguishing between 2ASK and 2PSK, but the direct phase standard deviation $\sigma_{dp}$ in Section 3.1 can be used to distinguish between them. Similarly, high-order cumulants are not effective in distinguishing between 2FSK and 4FSK, so spectral line features in the next section can be used for differentiation.

### 3.3. Feature Parameters Based on a Spectral Line

The mathematical model of the MFSK signal is as follows:(24)$$x(t)=\cos\left[2\pi f_{0}t+\sum_{n=-\infty }^{\infty }\sum_{m=1}^{M}\delta_{m}(n)\left[2\pi f_{m}(t-nT)+\theta_{m}(n)\right]q(t-nT)\right]$$

The mathematical model of the majority of FSK signals that are frequently utilized in practice is rewritten as follows:(25)$$x(t)=\sum_{n=-\infty }^{\infty }\sum_{m=1}^{M}\delta_{m}(n)q_{m}(t-nT)$$
(26)$$q_{m}(t)=\cos\left[2\pi \left(f_{0}+f_{m}\right)t+\phi_{m}\right]$$

The power spectral density function simulations of 2FSK and 4FSK signals are shown in Figure 4. The simulation parameters are as follows: for 2FSK modulation signal, the bit rate is 2.5 × 10^6^ Hz, the sampling frequency is 6 × 10^7^ Hz, and the carrier frequencies corresponding to the bits are 8.5 × 10^6^ Hz and 11.5 × 10^6^ Hz; for 4FSK modulation signal, the bit rate is 2.5 × 10^6^ Hz, the sampling frequency is 6 × 10^7^ Hz, and the carrier frequencies corresponding to the bits are 5.5 × 10^6^ Hz, 8.5 × 10^6^ Hz, 11.5 × 10^6^ Hz, and 14.5 × 10^6^ Hz.

When only the positive frequency portion is taken into account, Figure 4 shows that the power spectral density function of a 2FSK signal has two line spectra at a non-zero frequency, while that of a 4FSK signal has four line spectra. As a result, the feature parameter *N* can be considered to be the number of spectral lines in the signal, where *N = 2* and *N = 4,* respectively, represent 2FSK signals and 4FSK signals.

### 3.4. Feature Parameters Based on Cyclic Spectrum

As the carrier of the spectral correlation feature of modulated signals, the spectral correlation diagram usually exists in three dimensions, that is, the cyclic spectral correlation features of signals are distributed in a three-dimensional state. The generation of a spectral correlation map and the extraction of spectral correlation features require a large amount of computation, which is not conducive to the timeliness of modulation recognition algorithms.

By analyzing the cyclic spectral correlation function of the digital modulation signal, it is possible to determine that while the spectral correlation features are distributed in three dimensions, their primary features are concentrated on the α-section and the *f*-section, which are connected to the modulation type and parameters. The computational complexity can be decreased, and the resolution can be raised by examining and contrasting the cyclic spectral features on the α-section and the *f*-section. On the basis of this, the *R* cyclic spectrum feature parameter is designed in this study to extract the features of the signal. The following is the calculation expression:(27)R=[mean{f(n)}+mean{α(n)}]×number

In the formulas, mean{f(n)} is the average of the cross-section of cyclic spectrum f=0, mean{α(n)} is the average of the cross-section of cyclic spectrum α=0, and number is the number of significant peaks on the cross-section of cyclic spectrum α=0.

This study examines the numerical simulation analysis of the cyclic spectrum biaxial average’s feature parameter *R* in various signal-to-noise ratio situations. The sampling frequency is 10 kHz, the carrier frequency is 250 Hz, and the number of symbols is 350 for the 2ASK, 4ASK, 2PSK, and 4PSK signals, respectively. Setting parameters for a 2FSK signal include a sampling frequency of 10 kHz, a symbol count of 350, and carrier frequencies of 250 Hz and 750 Hz for each symbol. Setting parameters for a 4FSK signal include a sampling frequency of 10 kHz, a symbol count of 350, and carrier frequencies of 250 Hz, 750 Hz, 1250 Hz, and 1750 Hz for each symbol. Figure 5 displays the simulation results with the SNR environment set at −10 dB to 10 dB. As observed in the figure, the distinctive parameters *R* of the cyclic spectrum biaxial average of the other signals, with the exception of 2ASK, 2PSK, and 4PSK, are different, which can be useful in classifying and identifying them.

### 3.5. Feature Parameters Based on Autocorrelation

DSSS signals are commonly used in underwater communication due to their good concealment in underwater channels. DSSS signals consist of information sequences and pseudo-random sequences. Because the pseudo-random sequence has certain characteristics, it is possible to classify and identify direct sequence spread spectrum signals based on the regularity of the pseudo-random sequence itself. The commonly used method to determine the presence of spread spectrum signals is the second-order matrix test of the autocorrelation function. Because the second-order moment detection of the time domain correlation is stronger in suppressing noise than the time domain autocorrelation algorithm, making the periodic spectral peaks more prominent, this paper uses a time domain second-order moment detection algorithm to design the autocorrelation feature parameter S.

For simulation, set the following parameters: *N* = 255 for the length of the signal spread spectrum sequence; SNR = −10 dB, 10 for the number of signal codes; 7 for the order of the M sequence; 5 × 10^6^ Hz for the chip rate of the M sequence; 15 MHz for the carrier frequency; and 90 MHz for sampling. In Figure 6, which represents the second-order matrix of the autocorrelation function of the DSSS signal acquired through simulation, it is possible to see that there are clear periodic spectral peaks.

Figure 7 shows the autocorrelation simulation of various communication modulated signals. Depending on whether each signal’s autocorrelation function spectral line is periodic, the feature parameter *S* is designed accordingly. To distinguish a DSSS signal from other modulation signals, if *S* = 1, it is assumed that the signal’s autocorrelation function has periodic spectral lines, making it a DSSS signal. If *S* = 0, it is assumed that the signal’s autocorrelation function does not have periodic spectral lines, making it another modulation signal.

## 4. Simulation Results and Analysis

### 4.1. Simulation Environment and Results

To verify the feasibility and effectiveness of the proposed method, this paper conducted simulation experiments in the MATLAB environment. Seven commonly used underwater communication modulation signals, including 2ASK, 4ASK, 2FSK, 4FSK, 2PSK, 4PSK, and DSSS/BPSK, were selected, and $\sigma_{aa},\sigma_{af},\sigma_{ap},\sigma_{dp},F_{1},F_{2},F_{3},F_{4},N,R$, and *S* 11 feature parameters were combined with the AOA–RF algorithm for classification and identification.

The modulation signal parameters are set as follows: the bit rate is 500 Hz; the carrier frequency is 10 kHz; the sampling frequency is 100 kHz; and the direct sequence spread spectrum modulation method is BPSK, using a pseudo-random sequence with a period of 127 as the spreading sequence, with each chip carrying 8 carrier waves. The simulation channel is in an environment of AWGN. Using MATLAB software, 2000 data points are selected for each of the seven signals, and the feature parameters dataset is calculated for SNR ranging from −10 dB to 10 dB. Then, 7 × 11 × 500 data points are selected from the dataset as the test dataset to test the random forest algorithm. The recognition accuracy of each signal is shown in Figure 8.

From Figure 8, it can be seen that in the AWGN environment, the recognition rate for the above 7 modulation signals can reach 95% when SNR ≥ −5 dB. When SNR ≥ −3 dB, the recognition accuracy for the above 7 modulation signals can reach 100%. Among them, the recognition effect of the 2PSK signal is the best, and 100% recognition accuracy can be achieved when SNR ≥ −6 dB. The recognition effect of the DSSS signal is second, while the recognition effect of 2FSK and 4FSK signals is relatively poor.

### 4.2. Simulation Results under the BELLHOP Channel

To simulate the water acoustic environment as realistically as possible, this paper conducted simulation experiments based on the BELLHOP channel transmission overlay model at a distance of 2.5 km. The curves of the correct recognition rates of each modulation signal with the change of SNR are shown in Figure 9.

From Figure 9, it can be seen that in a more realistic underwater environment, the algorithm performs best in identifying the DSSS signals, with a recognition rate of over 95% at an SNR of −2 dB. In this environment, the previously best-performing 2PSK signal has a reduced recognition accuracy, which is lower than that of the DSSS, 2ASK, and 4ASK signals. As in the AWGN environment, the method has poor recognition performance for 4FSK and 2FSK signals.

To compare and verify the recognition performance of the algorithm under different channels, another set of BELLHOP channel models was simulated in this paper, namely the BELLHOP channel 5 km transmission superimposed model. The water surface is assumed to be a vacuum, and the seabed is flat. The specific parameters of the two sets of underwater acoustic channels are shown in Table 2. The curve of the average recognition rate of the signal set to be identified under different channel environments with the change of SNR is shown in Figure 10.

From Figure 10, it can be seen that in the AWGN channel, the recognition accuracy can reach 95% at −5 dB. BELLHOP channel models for 2.5 km and 5 km were created using the same ocean environment data, with the only difference in horizontal distance between the receiver and the sound source. As shown in the figure, with the increase in the transmission distance, the recognition accuracy decreases.

### 4.3. Comparison of Different Classifiers

This paper compares the overall recognition rates of the RF algorithm before optimization, the method based on high-order cumulants combined with RF, and the method based on instantaneous features combined with RF. The comparison is shown in Figure 11.

As shown in Figure 11, the proposed AOA–RF algorithm has significant advantages over the traditional RF algorithm, the RF algorithm based on instantaneous features, and the RF algorithm based on high-order cumulants. When SNR ≥ −4 dB, the average recognition rate of the signal set consisting of 7 modulation signals can reach 100%. The recognition rates of the RF algorithm based on instantaneous features and the RF algorithm based on high-order cumulants are close, and both reach 100% when SNR ≥ 1 dB. In the case of SNR, the average recognition rate of the AOA–RF algorithm for underwater communication modulation signals is 96.51%. Compared with the traditional RF algorithm, the RF algorithm based on instantaneous features, and the RF algorithm based on high-order cumulants, the recognition rates have been increased by 20.53%, 4.96%, and 6.60%, respectively.

Compared to the wavelet transform, BP neural network, and SVM classification recognition methods, the recognition accuracy is shown in Figure 12 and Table 3.

Based on Figure 12 and Table 3, it can be seen that the SVM method has a similar average recognition rate for the signal set compared to the method proposed in this paper. When SNR ≥ −4 dB, the recognition rate reaches 100%, but it drops to 98% when SNR = −3 dB, which also occurs in the BPNN method, leading to a decrease in the stability of the model. In addition, to ensure the accuracy of the classifier, the number of training samples used in the training process of BPNN and SVM is much larger than that of RF, which increases hardware burden and cost, and may not be feasible in underwater environments. Therefore, using the method proposed in this paper is a better choice because it requires fewer training samples, does not require too much hardware support, and is more suitable for underwater applications.

## 5. Conclusions

This paper proposes an RF classifier based on the AOA algorithm, analyzes the main feature parameters that affect signal recognition, fully utilizes the advantages of both the AOA and RF algorithms, and improves the accuracy and stability of signal recognition. This provides a new method for the modulation mode recognition of underwater acoustic signals.

This paper proposes 11 feature parameters based on the instantaneous characteristics, high-order cumulant features, spectral line features, cyclostationary features, and autocorrelation periodic spectral line features of each modulation signal for classification and identification. The simulation results show that when the SNR is higher than −7 dB, the recognition accuracy for 2ASK, 4ASK, 2FSK, 4FSK, 2PSK, 4PSK, and DSSS/BPSK can reach 80%. When the SNR is higher than −5 dB, the recognition rate for the above 7 modulation signals can reach 95%. When the SNR is higher than −3 dB, the recognition accuracy for the above 7 modulation signals can reach 100%. In addition, to simulate the real underwater environment, this paper also conducts simulations based on the BELLHOP underwater acoustic channel model, and the results show that this method still has good performance. Compared with wavelet transform, BPNN, and SVM, this method has the advantages of a simple implementation process, high recognition accuracy, and stable performance and can be better applied in underwater environments.

In the future, it is possible to explore more representative feature parameters as standards for identifying communication modulation signals, which may further improve recognition accuracy. In addition, the simulation test environment in this paper is mainly based on AWGN and the BELLHOP underwater acoustic channel model. In future work, it is possible to conduct actual sea trials to further demonstrate the reliability of this method.

## Figures and Tables

**Figure 1 sensors-23-02764-f001:**
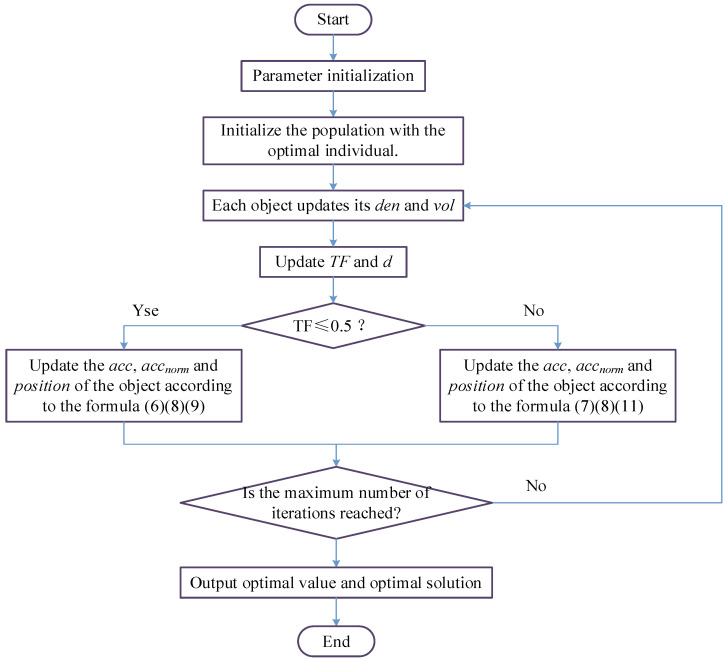
Flow chart of the AOA.

**Figure 2 sensors-23-02764-f002:**
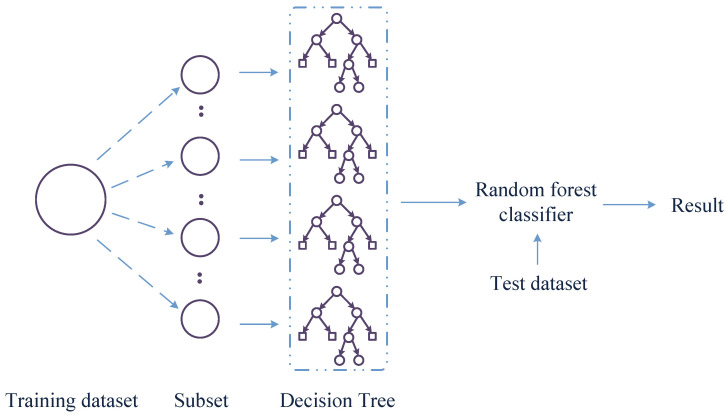
Flow chart of the RF.

**Figure 3 sensors-23-02764-f003:**
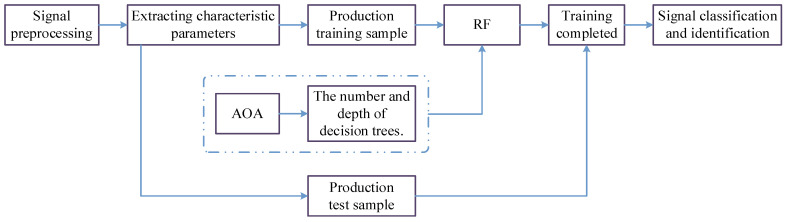
Overall technology roadmap for modulated signal identification.

**Figure 4 sensors-23-02764-f004:**
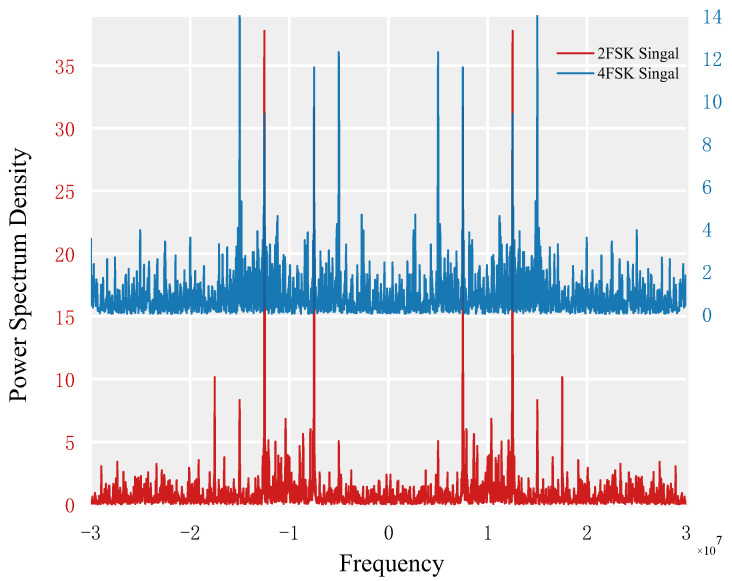
Power spectral density of 2FSK and 4FSK.

**Figure 5 sensors-23-02764-f005:**
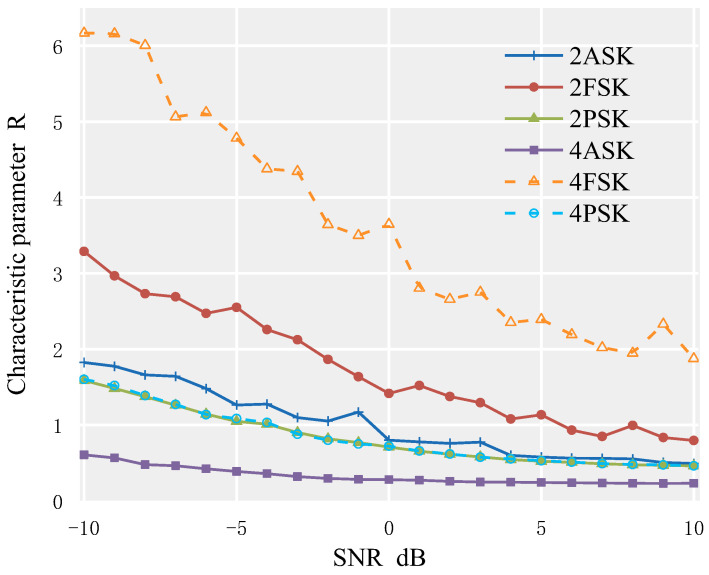
Cyclic characteristic parameter R of each modulated signal.

**Figure 6 sensors-23-02764-f006:**
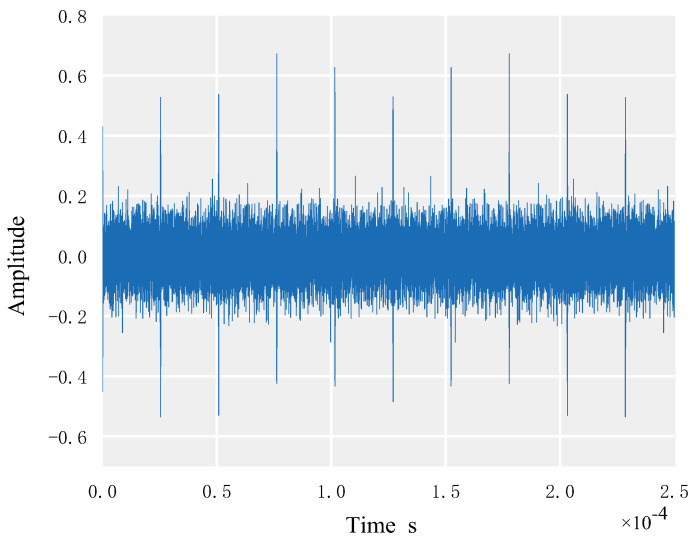
Second-order matrix of DSSS autocorrelation function.

**Figure 7 sensors-23-02764-f007:**
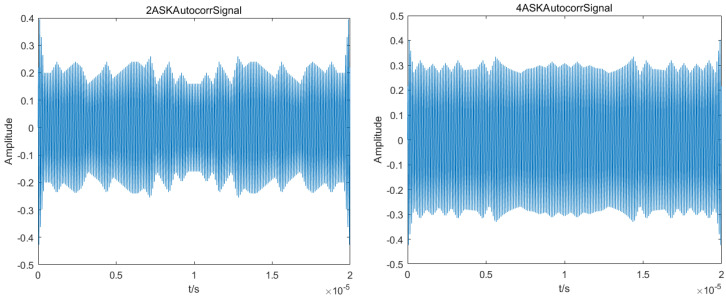
Autocorrelation function of each modulation signal.

**Figure 8 sensors-23-02764-f008:**
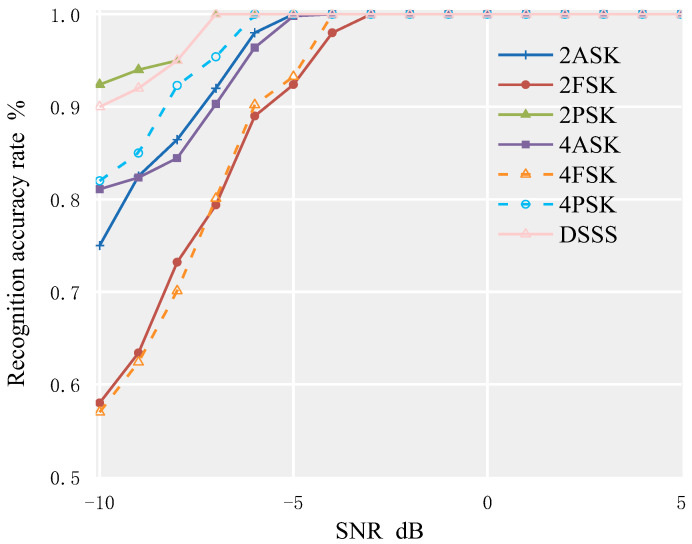
Recognition rate of seven modulation signals based on AOA–RF algorithm in AWGN environment.

**Figure 9 sensors-23-02764-f009:**
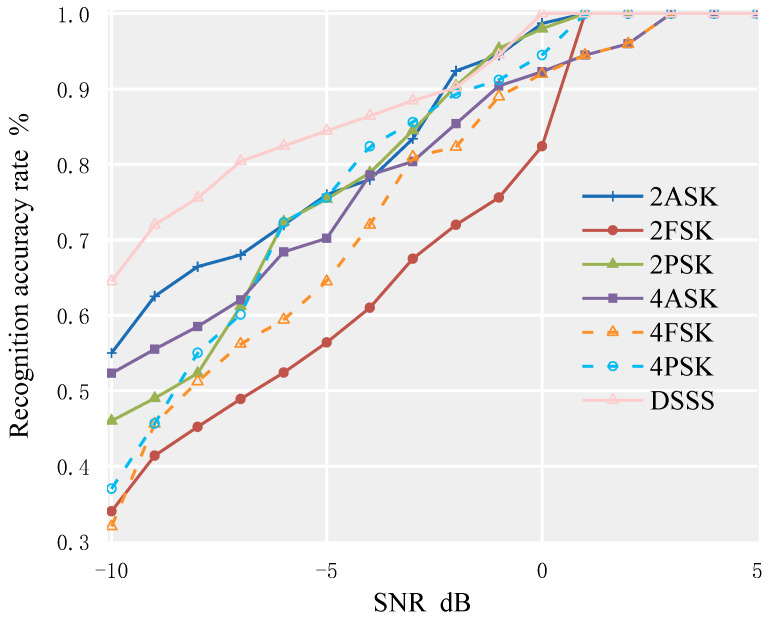
Accuracy of each signal identification under 2.5 km transmission superposition model of BELLHOP channel.

**Figure 10 sensors-23-02764-f010:**
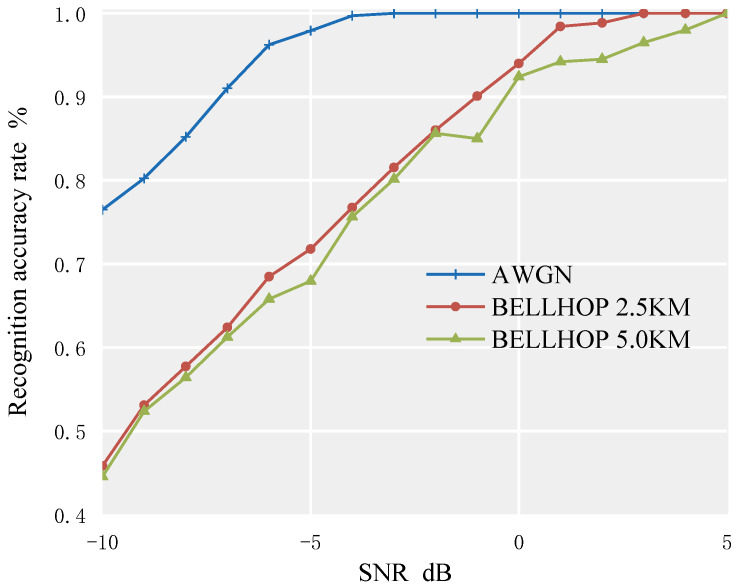
Comparison of recognition performance of AOA–RF algorithm in different channel environments.

**Figure 11 sensors-23-02764-f011:**
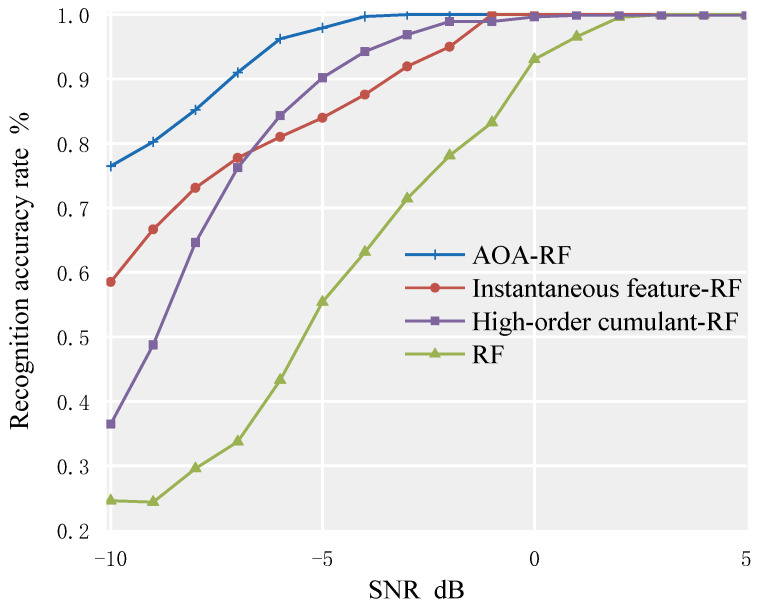
Comparison of average recognition rate between AOA–RF algorithm and traditional method.

**Figure 12 sensors-23-02764-f012:**
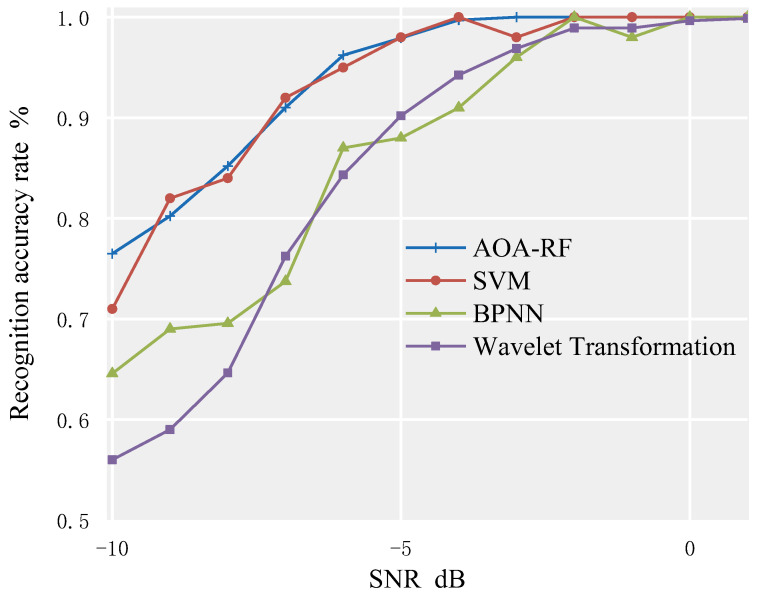
Comparison of average recognition rate between AOA–RF algorithm and new method.

**Table 1 sensors-23-02764-t001:** Theoretical values of feature parameters $F_{i}$ of six modulation signals.

Modulation Pattern	$F_{1}$	$F_{2}$	$F_{3}$	$F_{4}$
2ASK	1	1	2	32
4ASK	1	1	1.36	27.52
2FSK	0	0	1	16
4FSK	0	0	1	16
2PSK	1	1	2	32
4PSK	1	0	1	16

**Table 2 sensors-23-02764-t002:** BELLHOP channel parameters.

Parameter Name	Numerical Value (First Group)	Numerical Value (Group 2)
Signal frequency	10 khz	10 khz
Seawater depth	345.78 m	345.78 m
Sound source depth	100 m	100 m
Receiver depth	120 m	120 m
Number of receivers in the horizontal direction	50	50
Maximum transmission distance	2.5 km	5 km
Seawater density (kg/m^3^)	1024	1024
Sound speed of seawater (m/s)	1518	1518

**Table 3 sensors-23-02764-t003:** Recognition accuracy of different methods under different SNR environments.

Methods	Accuracy Recognition Rate in Different SNR Environments (%)					
	−10	−8	−6	−4	−2	0 (dB)
RF	24.57	29.57	43.29	63.14	78.14	93.04
Higher-order cumulant-RF	36.48	64.63	84.33	94.25	98.92	99.64
Wavelet Transform	56.00	64.63	84.33	94.25	98.92	99.64
Instantaneous feature-RF	59.12	73.11	81.02	87.59	100	100
BPNN	64.57	69.57	87.07	91.14	100	100
SVM	71.09	84.06	95.10	100	100	100
AOA-RF	76.50	85.21	96.23	99.71	100	100

## Data Availability

Not applicable.

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
