# Peer review of "Modulation Signal Recognition of Underwater Acoustic Communication Based on Archimedes Optimization Algorithm and Random Forest"

_sensors, 2023, doi:10.3390/s23052764_

Round 1

Reviewer 1 Report

Even though the paper presents an interesting issue, given the current format and the low number of references, it cannot be considered for publication. Nevertheless, the results seem promising and well-analyzed. Some general comments to help the authors include:

Improve the introduction by providing references about the importance of underwater communication modems and the lack of alternatives (10.1109/JSEN.2015.2434890 and 10.1007/s11277-020-07431-x)Add a related work (adding at least 10 to 15 references about similar published papers).

Check the conclusions and discussion content. Traditionally, conclusions are the last sections, and in the discussion, authors compare their results with existing similar solutions. Consider creating a comparative table in which the obtained results are compared with the results of related work. 

Reviewer 2 Report

In this study, seven modulation signals are selected as recognition targets and combined with feature parameters to improve underwater communication signal modulation pattern detection precision and the recognition performance of traditional signal classifiers. The underwater acoustic channel model was created using BELLHOP. The study further conducts simulated studies in the MATLAB environment to confirm the viability and efficacy of the different methodologies. This also conducts simulation experiments based on the 2.5km transmission superposition model of BELLHOP channel in an effort to replicate the real underwater acoustic environment as closely as possible. This study also simulates a group of BELLHOP channel models, including the 5km transmission superposition model of BELLHOP channel, in order to compare and evaluate the recognition performance of the method in various channels.

The article is an interesting study and is of current interest in the field of acoustics. This is a worthwhile piece of research in terms of technical aspects and addresses some important aspects in acoustic theory. The simulations seem to be valid and significant to understand the research. However, the article is poorly drafted, losing the actual essence of the investigation made in it. Therefore, I would suggest revising the article thoroughly thereby improving the presentation considerably. For example:

1.      The first sentence of abstract is incomplete. This shows a careless attitude while drafting the article. Further, the abstract lacks consistency and precision in terms of its contents. I would suggest to completely revise the abstract with a brief statement about the purpose and scope of the study, describe the research methods used and provide a summary of the results by keeping it concise and to the point, typically between 150 and 250 words. Also, use clear and concise language, avoiding technical jargon and abbreviations whenever possible and to be sure that the abstract accurately reflects the content of the article. Remember, the abstract is often the first thing readers will see, so it's important to make it engaging and informative.

2.      In keywords, adopt the uniform format setting, e, g; to Capitalize Each Word.

3.      At various parts of the article, it appears that the font and format are varied unnecessarily. It is important to use a consistent format and font size throughout the article.  

4.      Conclusion section is very poorly drafted. This sounds like an abstract. When writing the conclusion, avoid introducing new information or repeating information already covered in the article. Instead, focus on providing a concise summary of the study's key findings and their significance. Do discuss the implications and significance of the findings, including how they contribute to the existing knowledge in the field. Also, address any limitations or weaknesses of the study, and suggest areas for future research. Include an effective final statement that emphasizes the importance and relevance of the study, and potentially its broader implications.

5.      Proofread the whole article carefully for grammar and spelling errors. 

Round 2

Reviewer 1 Report

The authors have addressed my comments and the paper is ready to be accepted.

Reviewer 2 Report

I agree to the response provided by the authors in improving the quality of research article. Thus, the article can be accepted for publication in its present form.